# Anti-*Trypanosoma cruzi* Activity of Metabolism Modifier Compounds

**DOI:** 10.3390/ijms22020688

**Published:** 2021-01-12

**Authors:** Nieves Martinez-Peinado, Clara Martori, Nuria Cortes-Serra, Julian Sherman, Ana Rodriguez, Joaquim Gascon, Jordi Alberola, Maria-Jesus Pinazo, Alheli Rodriguez-Cortes, Julio Alonso-Padilla

**Affiliations:** 1Barcelona Institute for Global Health (ISGlobal), Hospital Clínic—University of Barcelona, 08036 Barcelona, Spain; nieves.martinez@isglobal.org (N.M.-P.); nuria.cortes@isglobal.org (N.C.-S.); quim.gascon@isglobal.org (J.G.); mariajesus.pinazo@isglobal.org (M.-J.P.); 2Department of Pharmacology, Toxicology, and Therapeutics, Veterinary Faculty, Autonomous University of Barcelona, 08193 Bellaterra, Spain; Clara.Martori@uab.cat (C.M.); Jordi.Alberola@uab.cat (J.A.); 3Department of Microbiology, New York University School of Medicine, New York, NY 10010, USA; Julian.Sherman@nyulangone.org (J.S.); Ana.RodriguezFernandez@nyulangone.org (A.R.)

**Keywords:** *Trypanosoma cruzi*, Chagas disease, metabolism drugs, phenotypic assays, cytotoxicity assays, chronic in vivo model, dorsomorphin, 17-DMAG

## Abstract

Chagas disease is caused by the protozoan parasite *Trypanosoma cruzi* and affects over 6 million people worldwide. Development of new drugs to treat this disease remains a priority since those currently available have variable efficacy and frequent adverse effects, especially during the long regimens required for treating the chronic stage of the disease. *T. cruzi* modulates the host cell-metabolism to accommodate the cell cytosol into a favorable growth environment and acquire nutrients for its multiplication. In this study we evaluated the specific anti-*T. cruzi* activity of nine bio-energetic modulator compounds. Notably, we identified that 17-DMAG, which targets the ATP-binding site of heat shock protein 90 (Hsp90), has a very high (sub-micromolar range) selective inhibition of the parasite growth. This inhibitory effect was also highly potent (IC_50_ = 0.27 μmol L^−1^) against the amastigote intracellular replicative stage of the parasite. Moreover, molecular docking results suggest that 17-DMAG may bind *T. cruzi* Hsp90 homologue Hsp83 with good affinity. Evaluation in a mouse model of chronic *T. cruzi* infection did not show parasite growth inhibition, highlighting the difficulties encountered when going from in vitro assays onto preclinical drug developmental stages.

## 1. Introduction

Chagas disease (or American Trypanosomiasis) is caused by the protozoan parasite *Trypanosoma cruzi* (*T. cruzi*). The disease was discovered more than 110 years ago, but it remains an important public health issue in Latin America. Moreover, the number of detected cases in nonendemic regions of Northern America and Europe has increased due to population movements during the last decades. Nowadays, it is estimated that over 6 million people are affected by the disease which causes more than 10,000 deaths per year [1].

*T. cruzi* infection clinically progresses in two well differentiated stages. First there is a short acute phase (4–8 weeks long) that is mostly asymptomatic and thus frequently undiagnosed and untreated. After these initial stages, the parasite enters into a persistent and long-lasting chronic phase that may remain asymptomatic or cause cardiac and/or digestive tissue damage in approximately 30% of patients, leading to years of disability and eventually death if left untreated [2,3].

Currently, there are two antiparasitic drugs available to treat *T. cruzi* infections: benznidazole (BNZ) and nifurtimox (NFX) [1]. Both are nitroheterocyclic derivatives with high efficacy against the acute stage of the infection. However, efficacy diminishes for both drugs against the chronic stage, which is when diagnosis is usually obtained [4]. The long term administration regimes to treat the disease have frequent side effects and results in low adherence to treatment [4,5,6,7]. Additionally, drug resistance has been observed in some discrete typing units (DTUs) [8], and no vaccine or chemo-prophylactic agents are available. Therefore, identification of safer and more efficacious drugs to treat the chronic stage of the infection is urgently needed [9].

Drugs targeting fundamental enzymes or metabolic pathways in the host cell could be a source of potential antiparasitic treatments, since metabolic coupling of intracellular pathogens with host cells is essential for successful colonization of the host [10]. In this respect, Caradonna and coworkers described that host metabolic networks of energy production, nucleotide metabolism, pteridine biosynthesis, and fatty acid oxidation are important for intracellular *T. cruzi* growth, providing experimental proof of the bond between host cells metabolic states and parasite development [10]. It was observed that host cells Coenzyme Q10 production was linked to pyrimidine biosynthesis and *T. cruzi* amastigotes replication [10]. Additionally, the serine/threonine kinase AKT, involved in the regulation of multiple cell proliferation and survival pathways, was identified as an important regulator of amastigotes replication, despite the mechanistic basis of its influence is yet unknown [10]. In general, a metabolic environment favoring fatty acid oxidation over glucose oxidation seemed to benefit intracellular amastigotes growth [10]. Similarly, *T. cruzi* replication was favored when cells maintained high ATP/ADP ratios, which could be induced by the inactivation of 5’-AMP-activated protein kinase (AMPK). AMPK controls cell energy homeostasis, inducing the switch between anabolic and catabolic pathways when ATP/ADP ratios are unbalanced [11].

In the search of novel chemical entities for Chagas disease, we evaluated a collection of compounds (AICAR, dorsomorphin, SC79, Akti-1/2, oligomycin, etomoxir, and 17-DMAG) and two licensed drugs (doxycycline and sodium salicylate), which target key metabolic enzymes modulating bioenergetic cellular pathways (Figure 1; chemical structures in Appendix A). AMPK is modulated positively by AICAR and sodium salicylate and negatively by dorsomorphin [12], while AKT is activated by SC79 and inhibited by Akti-1/2 [13]. Oligomycin and etomoxir have been, respectively, described to inhibit the mitochondrial ATP synthase and the carnitine palmitoyltranferase-1 (CPT-1). By inhibiting ATP synthase, oligomycin promotes the induction of glucose uptake through AMPK and AKT dual pathways, avoiding at the same time the intracellular uptake of calcium ions [14]. Despite its acknowledged toxicity we decided to include it as an inhibitor of mitochondria function. On the other side, etomoxir mediated inhibition of CPT-1 inhibits fatty acids oxidation [15]. Compound 17-DMAG was reported to inhibit the activity of the heat shock protein 90 (Hsp90) leading to protein misfolding and ubiquitin-dependent proteasome degradation [16]. It has also been described to antagonize hypoxia-inducible factor 1 alpha (HIF1α), a transcription factor that controls energy metabolism and angiogenesis under hypoxic conditions [17]. Doxycycline is a broad-spectrum antibiotic of the tetracycline family applied to the treatment of bacterial pneumonia, acne, early Lyme disease, gonorrhea, typhus, and syphilis [18]. It impairs synthesis of proteins encoded in the mitochondrial genome and changes gene expression patterns shifting metabolism towards a more glycolytic phenotype in human cells [19]. While the other licensed drug, sodium salicylate, activates AMPK, central regulator of cell growth and metabolism [20].

Herein, we explored whether aforementioned bioenergetic modulators specifically inhibited *T. cruzi* growth by means of antiparasitic and cell toxicity in vitro phenotypic assays. Binding of the most active one, 17-DMAG, to its main enzymatic target was further analyzed in silico. Ultimately, encouraged by retrieved results, 17-DMAG activity was also evaluated in an in vivo model of chronic *T. cruzi* infection.

## 2. Results

### 2.1. Anti-T. cruzi Activity

Compound activity against *T. cruzi* mammalian stages was determined using a phenotypic assay based on African green monkey kidney epithelial cells (Vero, ATCC^®^ CCL-81™) infected with an engineered *T. cruzi*—Tulahuen strain (DTU VI) expressing beta-galactosidase [21,22]. In every run of the assay we included the reference drug BNZ as control. This had an average IC_50_ value of 1.63 (0.11) μmol L^−1^ (Figure 2) which is similar to previous reports [22,23].

Upon evaluating the nine compounds in at least three independent experiments, six of them were found to be inactive against the parasite (AICAR, SC79, Akti-1/2, etomoxir, sodium salicylate, and doxycycline; Appendix A). In contrast, dorsomorphin, oligomycin, and 17-DMAG showed a *T. cruzi* growth inhibition activity more potent than that of the reference drug BNZ (Figure 2 and Appendix A). Respectively, dorsomorphin, oligomycin, and 17-DMAG yielded average IC_50_ values of 0.24 (0.021) µmol L^−1^, 0.52 (0.047) µmol L^−1^, and 0.017 (0.001) µmol L^−1^ (Table 1).

### 2.2. Identification of Compounds with Specific Activity against the Parasite

A secondary cell toxicity assay with monkey Vero cells was performed to test for parasite specificity of dorsomorphin and 17-DMAG compared to host cells. Oligomycin was discarded from further evaluation because it is highly toxic to human subjects [24].

Since compounds under study were metabolic inhibitors, we evaluated their impact on host cells viability with two different methodologies: AlamarBlue and crystal violet assays (see Section 4.5 for details). BNZ, included as reference in the Vero toxicity assays, yielded an average TC_50_ value of 243.8 (17.04) µmol L^−1^ in the AlamarBlue-based assay and 140.2 (13.18) µmol L^−1^ in the crystal violet assay as reported (Table 1, Appendix A) [22,23]. In order to consider any of the two compounds suitable for further progression we determined a selectivity index (SI, i.e., the ratio between TC_50_ and IC_50_) > 10 in both readouts. Such SI ratio is commonly used to consider compounds with specific antiparasitic activity [25].

Vero cells toxicity of dorsomorphin in the AlamarBlue fluorescent-based readout [TC_50_ = 16.6 (1.88) µmol L^−1^] indicated that its activity against the parasite was specific (SI = 69.2; Table 1, Figure 3). However, the colorimetric readout revealed a higher toxicity profile for Vero [TC_50_ = 0.26 (0.032) µmol L^−1^] which resulted in a SI value < 10 (Table 1, Figure 3).

17-DMAG had an average TC_50_ value of 6.23 (0.86) µmol L^−1^ against Vero cells in the AlamarBlue assay (SI = 366.5; Table 1, Figure 3). Then, although crystal violet readout resulted again in higher toxicity [TC_50_ = 2.97 (0.37) µmol L^−1^] (Table 1, Figure 3), its SI prevailed > 10 (SI = 174.7). In fact, 17-DMAG showed wider selectivity indexes than those obtained with the reference drug BNZ (Table 1).

### 2.3. Anti-Amastigote Specific Activity of 17-DMAG

Amastigotes are considered the main target for any prospective drug to treat chronic *T. cruzi* infections. Since 17-DMAG activity was determined to be anti-parasite specific, we assessed its anti-amastigote activity by means of a biological assay specifically targeting the mammalian replicative form of the parasite. In this anti-amastigote assay, pre-infected cells from which trypomastigotes had been washed away were exposed to the compounds after allowing the intracellular amastigotes 18 h to settle. The standard drug BNZ was included for comparison. Overall, we found that the anti-amastigote activity of 17-DMAG was ≈15× less potent than what we had retrieved in the antiparasitic assay previously described: IC_50_ = 0.27 (0.017) µmol L^−1^ versus IC_50_ = 0.017 (0.001) µmol L^−1^ (Table 1, Figure 4). Anti-amastigote IC_50_ of BNZ was 2.02 (0.08) µmol L^−1^ versus 1.63 (0.11) µmol L^−1^. Despite this reduction in the activity levels of 17-DMAG against the amastigote forms, its SI window over Vero cells remained >10 (Table 1).

### 2.4. Hsp90 Is the Target of 17-DMAG

We further inquired if the observed antiparasitic effect of 17-DMAG could be related to the inhibition of the mammalian Hsp90 or if the compound could also be targeting *T. cruzi* Hsp90 homologue Hsp83 [26]. Previous studies have described that 17-DMAG binds with higher affinity to the N-terminal domain of Hsp83 proteins from *T. cruzi* close relatives *Trypanosoma brucei* and *Leishmania amazonensis*, as well as to that of *Plasmodium falciparum* in comparison to human Hsp90 [27,28,29,30]. Multiple sequence alignment showed a high percentage of identity (>60%) between human Hsp90 and its parasite homologues [31] (Appendix A). Particularly, identity of N-term domains, where the ATP-binding pocket is located, was 73% between human Hsp90 and *T. cruzi* Hsp83 (Appendix A).

Since the 3D structure of human Hsp90 bound to 17-DMAG is available [32], we performed a morphological analysis of the interaction between this compound and *T. cruzi* Hsp83 protein with PyMol [33]. It revealed a different surface conformation of the ATP-binding pocket between both chaperones (Figure 5). Alignment-driven in silico mutagenesis entailed substituting human Hsp90 lysine (K) 112 for an arginine (R) 97 residue in *T. cruzi* Hsp83. In Hsp90 protein, the amine group of L112 forms a hydrogen bond of 3 Å with the C-18 ketone of 17-DMAG [32]. In contrast, the two amine groups of R97 in *T. cruzi* Hsp83 would interact with the C-18 ketone upon the formation of two bonds of 2.8 and 2.9 Å (Figure 5). Substitution of isoleucine (I) 91 onto a valine (V) did not seem to affect the water-mediated interaction.

In addition to the visualization of 17-DMAG interaction with the ATP-binding pockets of Hsp90 and Hsp83, we evaluated the binding affinity of 17-DMAG to both chaperones with Vina Wizard software [34]. This analysis predicted a free energy value of −9.7 Kcal/mol for the binding of 17-DMAG to Hsp90, whereas it was −6.2 Kcal/mol for *T. cruzi* Hsp83.

### 2.5. In Vivo Anti-T. cruzi Activity of 17-DMAG in a Mouse Model of Chronic Infection

Ultimately, antiparasitic activity of 17-DMAG was assessed in an in vivo model of chronic *T. cruzi* infection with a transgenic parasite expressing luciferase (Brazil strain, DTU I) [35,36]. In this assay, the signal provided by luciferase after injection of its substrate in the animals is proportional to the *T. cruzi* load, and it is considered a surrogate for parasitemia. Infected mice reached the peak of parasitemia at day 23 and treatment started at day 132 once chronic infection had been established (Figure 6A). One experimental group was treated intraperitoneally with 30 mg/kg/day of 17-DMAG for five days (Figure 6A,B), while other was treated five times every three days (over a 15 days period) with same dose (Figure 6A,C). We used these dosage and regimes following previous works by Meyer and coworkers and Santos and coworkers, respectively [37,38]. Vehicle-treated and BNZ-treated groups (same regimes and dose: 30 mg/kg/day) were included as controls. None of the groups treated with 17-DMAG showed statistically significant differences versus either vehicle or BNZ groups. Similarly, no statistically significant differences in the registered luciferase signal were observed between daily or intermittently treated mice, either with 17-DMAG or BNZ (data not shown). After treatment, mice treated daily and those treated intermittently with 17-DMAG, respectively had mean luminescence values (AU) of 1.95 × 10^6^ (±5.60 × 10^5^) and 8.69 × 10^6^ (±1.27 × 10^7^) photons s^−1^. Such values were higher than the 5 × 10^5^ photons s^−1^ threshold indicative of parasite clearance that had been determined by imaging a noninfected mouse injected with luciferase substrate the same as infected [39].

At the dose used (30 mg/kg/day), BNZ did not manage to completely clear the infection. Nonetheless, in the group of animals that received the drug intermittently over 15 days, there was a 6-fold significant reduction of parasitemia after treatment (*p* < 0.01, comparing levels at days 132 and 147) (Figure 6C). Finally, it must be noted that mice under the 17-DMAG daily regime showed signs of treatment associated toxicity. These were not observed in the animals that were intermittently treated with the compound. None of the animals treated with BNZ showed signs of drug-associated toxicity.

## 3. Discussion

Out of the nine metabolic inhibitors evaluated, three showed a highly potent antiparasitic activity: dorsomorphin, 17-DMAG, and oligomycin. The latter was included to test that a functional host cell mitochondria was important for the parasite survival, but it was discarded from further progression for its reported acute toxicity.

Cell toxicity assessment of the other two active compounds revealed that only 17-DMAG was specific against the parasite. This was concluded upon performing two complementary cell toxicity assays with AlamarBlue and crystal violet substrates [40,41], which measure cell metabolic activity and membrane integrity, respectively. No significant differences in 17-DMAG TC_50_ values were recorded, and the fact that toxicity was slightly higher in the crystal violet assay might have been due to its distinct readout addressing membranes integrity.

Dorsomorphin (also known as compound C) was originally identified as an AMPK inhibitor and widely used in cancer studies [11]. In this study it showed a high antiparasitic activity in the anti-*T. cruzi* assay, but it was also shown to be highly toxic against Vero cells [TC_50_ = 0.26 (0.032) µmol L^−1^] in the crystal violet based cell toxicity assay. Flagged as nonspecific against the parasite, it was discarded from subsequent analysis.

Regarding 17-DMAG, to the best of our knowledge, this is the first time that its anti-*T. cruzi* growth inhibition activity has been described. It was highly potent, in the submicromolar range (IC_50_ = 17 nmol L^−1^), in the primary antiparasitic assay. Moreover, its activity was determined to be parasite specific also in the anti-amastigote assay. That 17-DMAG potency was lower in the latter might be due to amastigotes being less accessible to it as intracellular forms. Nonetheless, still a strong submicromolar antiparasitic activity was registered (IC_50_ = 0.17 μmol L^−1^).

17-DMAG is a derivative of geldanamycin, which has been reported to inhibit the activity of the Hsp90 protein family [16], and to antagonize HIF1α transcription factor [17] (Figure 1). Amongst different Hps90 inhibitors like geldanamycin, 17-AAG or 17-DMAG, the latter has been chosen in multiple assays due to its potency, selectivity, and better water solubility [37]. Hsp90 proteins are chaperones modulating availability of client proteins (kinases and nuclear receptors) thus enabling important cellular processes such as cell growth, signal transduction, and development [29]. Eukaryotic cells have three types of Hsp90s: cytosolic Hsp90 with two isoforms (Hsp90α and Hsp90β), Grp94 (glucose-regulated protein 94) at the endoplasmic reticulum (ER), and the mitochondrial Trap1 (tumor necrosis receptor-associated protein 1) [16]. The sequence of Hsp90 proteins is rather conserved and its activity has been documented also in intracellular protozoan parasites such as those within genus *Trypanosoma*, *Leishmania*, *Toxoplasma*, and *Plasmodium* [37,42]. Three functional domains are found within Hsp90s: a N-term domain with an ATP-binding site and ATPase activity; a middle region where the binding site for client proteins is located; and a homo-dimerization domain with another binding site for ATP at their C-term [16] (Figure 1; Appendix A). Remarkably, structural studies have shown that compound 17-DMAG binds to the ATP-binding site of the N-term domain [32]. Others have investigated the use of 17-DMAG against various human pathogens [28,30,37,43]. Palma and coworkers reported a higher selectivity of 17-DMAG against *L. amazonensis* cells (IC_50_ = 86.1 nmol L^−1^) than to THP−1 host cells (TC_50_ = 10.6 µmol L^−1^) [28]. Similarly, Meyer and coworkers demonstrated its higher selectivity against *T. brucei* (IC_50_ = 3.1 nmol L^−1^) in comparison to L1210 mouse cells (TC_50_ = 1000 nmol L^−1^) [37], and *P. falciparum* growth was also suppressed by 17-DMAG at nontoxic concentrations to the host cells [30]. Notably, in vivo studies have already shown that 17-DMAG, administered either intraperitoneally or orally at a supposedly nontoxic dose of 30 mg kg^−1^ for five days cured mice of a lethal *T. brucei* infection decreasing the parasitemia to the limit of detection by day three [37].

Besides, 17-DMAG has successfully reached phase I clinical studies in patients with advanced solid cancers [44,45,46] and lymphocytic and myeloid leukemia [47]. The highest dose reached in these phase I studies without experimenting dose-limiting toxicity was 80 mg/m^2^, even though lower doses were also defined as effective [45,46]. Most common adverse events were fatigue, nausea, vomiting, anorexia, gastrointestinal, liver function changes, and ocular alterations [44,45]. The maximum blood tolerated concentrations were 600–2700 nmol L^−1^ [37], which would exceed the 17–270 nmol L^−1^ IC_50_ range we determined against *T. cruzi* in this work.

*T. cruzi* Hsp90 homologue Hsp83 participates in the parasite cell cycle by controlling the differentiation of insect to mammalian infective forms [26]. It has also been described to have high ATPase activity, particularly stimulated in the presence of peptides [26]. Since a 73% sequence conservation was observed between N-term of *T. cruzi* Hsp83 and human Hsp90 [48], we decided to follow an approach to visualize a hypothetical Hsp83–17-DMAG interaction. Somewhat, in silico results unveiled a more embracing binding pocket surface to 17-DMAG in the modeled *T. cruzi* Hsp83 in comparison to human Hsp90 (Figure 5). The model also showed that the unique interaction of 17-DMAG C-18 ketone with human Hsp90 K112 was substituted by two bonds with *T. cruzi* Hsp83 R97, which would have an expected shorter binding distance from 3Å to 2.8 and 2.9 Å, respectively. Such differences may allow a swifter binding of 17-DMAG to Hsp83 N-term domain. In other related parasites, Palma and coworkers reported a higher affinity of 17-DMAG to *L. amazonensis* Hsp83 than to human Hsp90 based on in vitro assays and docking-structural analysis [28]. Additionally, thermal shift assays showed an affinity up to two-fold higher for *T. brucei* Hsp83 than to human Hsp90 [28]. However, the predicted energy required to disrupt 17-DMAG binding to *T. cruzi* Hsp83 was lower than that predicted for the human Hsp90 (ΔG = −6.2 Kcal/mol (−25.9 KJ/mol), versus ΔG = −9.7 Kcal/mol (−40.6 KJ/mol)). This would indicate a lower binding affinity between 17-DMAG and Hsp83. The predicted binding affinity value of 17-DMAG to the human Hsp90 correlates with that previously reported (ΔG = −32.3 KJ/mol) [49]. Our Vina Hsp83 model did not take into consideration the presence of solvent in the interface and the conformational changes that occur during the complex formation. It has been described that water-mediated hydrogen bonds between Hsp90 inhibitor compounds and amino acid residues within N-term domain of *T. brucei* Hsp83 were important as they simulated the positioning of one of the phosphate groups of ATP [27]. Furthermore, Nilapwar and coworkers reported the importance of water-mediated hydrogen bonds by direct bridging or by forming a network of water molecules in human Hsp90 complex formation with geldanamycin, 17-AAG and 17-DMAG [49], and it is also known that conformational changes coupled to ligand binding can contribute to modify the binding affinity [49]. Whether 17-DMAG in vitro antiparasitic effect is due to an inhibition of the host cells Hsp90 or *T. cruzi* Hsp83 remains to be elucidated. Moreover, since 17-DMAG has also been described to inhibit host cell glycolysis route through HIF1α, a change in the metabolic state of the cells might as well be having an impact in the parasite intracellular multiplication.

The potent and specific activity of 17-DMAG against *T. cruzi* in vitro, and its formerly described in vivo activity against closely related parasites [37,38], prompted its evaluation in a chronic model of *T. cruzi* infection. In this, 17-DMAG did not inhibit parasite presence in the infected mice. We did not observe a statistically significant decrease in parasitemia in mice treated intraperitoneally with 17-DMAG at 30 mg/kg daily for five days or those intermittently treated every three days over 15 days (five times in total) in comparison with vehicle inoculated animals (Figure 6). Neither was that presence eliminated by similar treatment (dosage and regime) with Chagas disease standard drug BNZ. Nonetheless, decay in parasitemia was indeed observed in those animals intermittently treated with BNZ, likely due to longer availability of the drug to act and contribute to reduce parasite presence. Clearance of *T. cruzi* parasitemia with BNZ in chronically infected mice has been described upon administration of ≥100 mg/kg/day over at least 20 days [50,51,52]. Reason to include a control group of animals treated with 30 mg/kg/day was to parallel test groups in regime and dose.

Administering 17-DMAG at a higher dose than the one evaluated here may not be an option in future experiments, at least in its daily administration regime and in this mouse strain. Among the observed signs of toxicity in mice within daily treated group, there were lethargy and ruffled fur. Formerly, Meyer and coworkers did not report signs of toxicity in *T. brucei* infected mice treated with the same dose [37]. This could be explained by the use of different mouse strains [53], as well as by the immunological compromise caused by each parasite when treatment started. Herein reported failure of 17-DMAG in a chronic in vivo model reminds that of posaconazole, which also presented a very potent in vitro anti-*T. cruzi* activity [54]. BNZ and NFX remain as the best possible Chagas disease treatments at the moment. It will be fundamental to determine whether alternative regimes currently under clinical evaluation manage to reduce their frequent adverse effects while maintaining (or increasing) efficacy [9]. Until then, efforts to try to identify a safe and highly effective therapy should continue.

## 4. Materials and Methods

### 4.1. Collection of Compounds/Drugs

Sodium salicylate (CAS N 54-21-7), oligomycin from *Streptomyces diastatochromogenes* (CAS N 1404-19-9) and doxycycline monohydrate (CAS N 17086-28-1) were purchased from Sigma-Aldrich (St. Louis, MO, USA). 17-DMAG (CAS N150270-08-9) was purchased from Tebu-bio (Barcelona, Spain). AICAR (CAS N 2627-69-2), Akti-1/2 (CAS N 612847-09-3), dorsomorphin dihydrochloride (CAS N 1219168-18-9), SC79 (CAS N 305834-79-1), and (R)-(+)-etomoxir sodium salt (CAS N 828934-41-4) were purchased from TOCRIS (Bristol, UK).

AICAR, Akti-1/2, dorsomorphin, etomoxir, and SC79 stock dilutions were prepared in DMSO at 20 mM whereas oligomycin and 17-DMAG were prepared at 2 and 1 mmol L^−1^, respectively. The drug sodium salicylate was prepared before use in the appropriate culture media at a starting concentration of 200 µmol L^−1^.

### 4.2. Host Cells Cultures

Vero (green monkey kidney epithelial cells) [55], LLC-MK2 (Rhesus monkey kidney epithelial cells) [22] were kept with DMEM supplemented with 1% penicillin-streptomycin (100 units/mL of penicillin and 100 μg/mL of streptomycin; P-S) and 10% heat inactivated fetal bovine serum (FBS) at 37 °C, 5% CO_2_, and >95% humidity as described [21].

### 4.3. Culture of T. cruzi Parasites

*T. cruzi* parasites from the Tulahuen strain (Discrete Typing Unit VI) expressing β-galactosidase were kindly provided by Dr. Fred Buckner (University of Washington, Seattle, WA, USA) and maintained using LLC-MK2 cells as hosts in DMEM supplemented with 2% FBS and 1% P-S-G [21]. Free-swimming trypomastigotes were purified as described to keep the parasite cycle in LLC-MK2 cells and for the performance of the antiparasitic assays [22].

### 4.4. Assay to Detect T. cruzi Growth Inhibition in 96-Well Plates

Our assay is based on Vero cells as hosts and infective trypomastigotes from Tulahuen strain that express the bacterial β-galactosidase enzyme as reporter activity [21]. Firstly, compounds were added in the first column of a 96-well tissue culture treated plates at an initial concentration of 200, 20, or 10 µmol L^−1^ and diluted in assay medium following either a 1:2 or 1:3 fold pattern to conform dose–response plate-maps. Vero cells and purified trypomastigotes were harvested, counted, and diluted to a concentration of 1 × 10^6^ cells/mL each. Then these were mixed and 100 μL added per well (50,000 Vero cells and 50,000 trypomastigote cells per well; multiplicity of infection or MOI = 1) [56]. The reference drug BNZ was used as control of drug growth inhibition in each run launched, whereas each plate contained its own negative and positive control as previously described [21,56]. After 4 days at 37 °C, 50 µL of a PBS solution containing 0.25% NP40 and 500 µM chlorophenol red-β-d-galactoside (CPRG) substrate were added per well [22]. Plates were further incubated at 37 °C for another 4 h and the absorbance read out at 590 nm using an Epoch Gene5 spectrophotometer. All experiments were performed at least in triplicate.

Additionally, based on Vero cells as hosts and the recombinant *T. cruzi* strain expressing β-galactosidase, we further adapted the antiparasitic assay described above to determine whether the activity of the compound 17-DMAG was specific against the intracellular amastigote forms. For this, Vero cells were seeded in T-175 flasks (5 × 10^6^ cells/flask) in DMEM supplemented with 1% penicillin-streptomycin and 10% FBS and cultured for 24 h. Then, cells were washed once with PBS and free swimming trypomastigotes (1 × 10^7^ trypomastigotes per flask; MOI = 1) in assay medium were added and allowed 18 h to infect. Infected cell monolayers were washed with PBS and detached from the flasks [23]. Cells were counted and diluted to a concentration of 5 × 10^5^ cells per mL, before adding 100 µL per well to test plates already containing the drugs as previously described. Each plate contained the same controls as previously described. Test plates were incubated for 72 h and their read out was performed as described above.

### 4.5. Cell Toxicity Assays

In this case compounds were added to tissue culture treated 96-well plates following a dose–response dilution pattern 1:2 or 1:3 with starting concentrations of 800 µmol L^−1^ for dorsomorphin and 64.8 µmol L^−1^ for 17-DMAG. Cell viability was checked upon cell counting with Trypan blue staining and we only proceeded if it was >95%. Vero cells were diluted at a concentration of 5 × 10^5^ cells per mL, before adding 100 µL per well. Each run contained its own negative and positive controls as described. Plates were incubated at 37 °C for 4 days and we used two different readout methodologies. One was based on the traditional AlamarBlue substrate (resazurin-based), which provides a quantitative measure of cells viability by their capacity to reduce resazurin into resorufin [40]. However, given that inhibitors may impact cells redox state, a distorted toxicity profile could have been obtained, and a parallel readout relying on crystal violet was also used. Vero cells grow as monolayers, and adherent cells detach from the tissue culture plate upon cell death. Thus crystal violet dye would only bind to structures in living cells [41]. In the case of AlamarBlue assays, we added 50 µL per well of a PBS solution containing 10% AlamarBlue reagent (ThermoFischer, Waltham, MA, USA) and incubated the plates for 6 h at 37 °C before reading the fluorescence intensity in a Tecan Infinite M Nano+ reader (excitation: 530 nm, emission: 590 nm). In the case of crystal violet assays, cells were washed with PBS and fixed with 100 µL of 10% formol for 15 min. Then, after washing again we added 50 µL of a crystal violet dilution (1:5 *v*/*v*) prepared from the stock and left it for 10 min. Crystal violet stock solution contained 250 mg of crystal violet powder (Sigma-Aldrich), 1 mL of EtOH, and 49 mL of H_2_Od. After that, crystal violet excess was eliminated washing with H_2_Od two times and plates were left to dry in the flow cabinet. Finally, we added 50 µL of SDS 2% for 30 min and read out absorbance at 590 nm in an Epoch Gene5 spectrophotometer. All experiments were performed at least in triplicate.

### 4.6. In Silico Study of 17-DMAG Binding to Hsp90/83 N-Terminal Domains

Hsp90 N-term domain was visualized with PyMol software (version 4.4.0) [33]. We used PyMol as well to visualize *T. cruzi* Hsp83 N-term domain by means of an alignment-directed mutagenesis upon the N-term domain of the cocrystalized human Hsp90 and 17-DMAG complex (PDB: 1OSF) [32]. We used BLAST [31] to identify the residues that differ between human Hsp90α isoform 2 (NCBI ref.: NP_005339.3) and Hsp83 of different *T. cruzi* strains. These were obtained from NCBI and TriTrypDB [57,58]. We chose *T. cruzi* Hsp83 from Dm28c strain as representative (NCBI ref.: PBJ77530.1) for its visualization in PyMol, but the same identity was shown between Hsp83 from other *T. cruzi* strains and human Hsp90 (Appendix A).

We used Vina Wizard software (on PyRx-virtual screening tool) [34] to calculate the binding affinity of 17-DMAG to Hsp90/83. The structure of 17-DMAG bound to the N-term domain of human Hsp90 was obtained from the cocrystalized complex. On the other hand, the N-term domain of *T. cruzi* Hsp83 was modeled using the SWISS-MODEL homology-modeling tool [59] by inputting the *T. cruzi* Dm28c amino acid sequence. This model was constructed based on *T. brucei* Hsp83 3D-structure available from the RCSB Protein Data Bank (PDB: 3O6O). These Hsp83 proteins shared a 90% identity, which was 94% between their N-term domains.

### 4.7. T. cruzi Chronic In Vivo Assay

Trypomastigotes from transgenic *T. cruzi* Brazil strain (DTU I) expressing firefly luciferase were purified, diluted in PBS, and injected in Balb/c mice (10^3^ trypomastigotes per mouse) [39]. Parasites from Brazil strain have been described to be highly virulent and susceptible to BNZ [60]. At defined time points, mice were anesthetized by inhalation of isoflurane and injected with 150 mg/kg of D-Luciferin Potassium-salt (Goldbio, St. Louis, MO, USA) dissolved in PBS [35]. Mice were imaged 5–10 min after injection of luciferin with an IVIS 100 (Xenogen, Alameda, CA, USA), and the data acquisition and analysis were performed with the software LivingImage (Xenogen). On day 132 post-infection, treatment was started by intraperitoneal administration of 17-DMAG at 30 mg/kg. Two schedules of treatment were followed: one experimental group was treated daily for five days [37]; the other was treated five times over 15 days [38]. Each group had its positive control group which received a similar dose of 30 mg/kg of BNZ on corresponding days. Negative control group received only vehicle solution [35]. Each group was formed by five mice. On the scheduled days 7, 14, 23 132, 141, and 147 mice were imaged after anesthesia and injection of luciferin as described above.

### 4.8. Data Analysis

Absorbance and fluorescence values respectively derived from the anti-*T. cruzi* and cell toxicity assays were normalized to the controls [56]. IC_50_ and TC_50_ values were determined with GraphPad Prism 7 software (version 7.00, 2016) using a nonlinear regression analysis model defined by Equation (1):(1)$$Y=100\;÷\left(\left(1+X^{HillSlope}\right)÷\left(IC50^{HillSlope}\right)\right)$$

IC_50_ and TC_50_ values are, respectively, defined as the compound concentration that inhibits parasite growth by 50% or is toxic to 50% of cells. Values provided are means and standard deviation [mean (SD)] of at least three independent experiments. Similarly, total flux values recorded in the in vivo experiments are shown as mean and SD. We used ANOVA with Bonferroni’s correction for multiple comparisons to determine if the differences between groups were statistically significant.

Comparison of luciferase activity levels in *T. cruzi*-infected mice after treatment was performed between days 132 (day of treatment) and days 141 (for 5 days treatment) or 147 (for 15 days intermittent treatment) by Mann–Whitney test.

## 5. Conclusions

Upon evaluation of nine metabolic modifier compounds we identified one with selective anti-*T. cruzi* activity in the sub-micromolar range: 17-DMAG. It had potent IC_50_ (0.017 µmol s^−1^) and good SI rates (SI = 366.5 and 174.7, for the AlamarBlue and crystal violet Vero assays, respectively). In addition, the in silico directed mutagenesis predicted a hypothetical better binding to *T. cruzi* Hsp83. Regrettably, 17-DMAG did not present parasite inhibitory activity in an in vivo chronic model of *T. cruzi* infection.

## Figures and Tables

**Figure 1 ijms-22-00688-f001:**
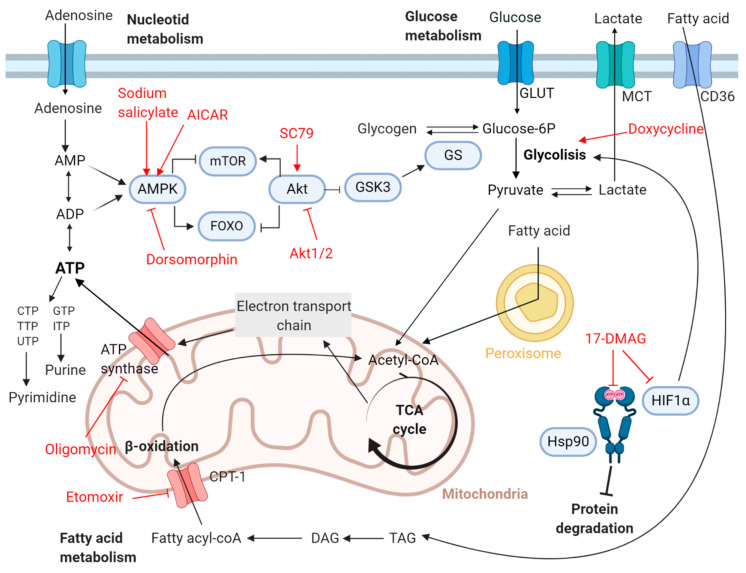
Host Cell Metabolic Routes Scheme. It shows known targets and effects on them of the compounds evaluated in this work. (Figure created with Biorender.com).

**Figure 2 ijms-22-00688-f002:**
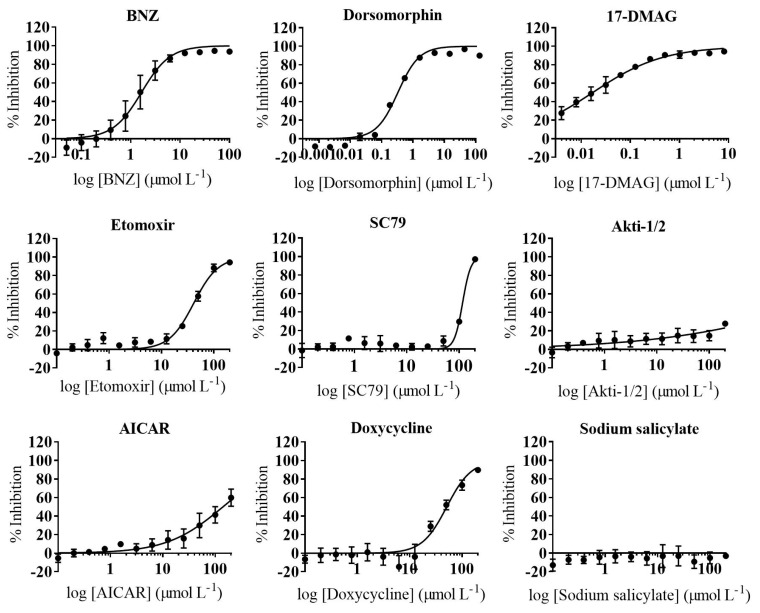
Anti-*Trypanosoma cruzi* Assay Dose–Response Curves. Graphs represent mean results and standard deviation (SD) of at least three independent biological replicas.

**Figure 3 ijms-22-00688-f003:**
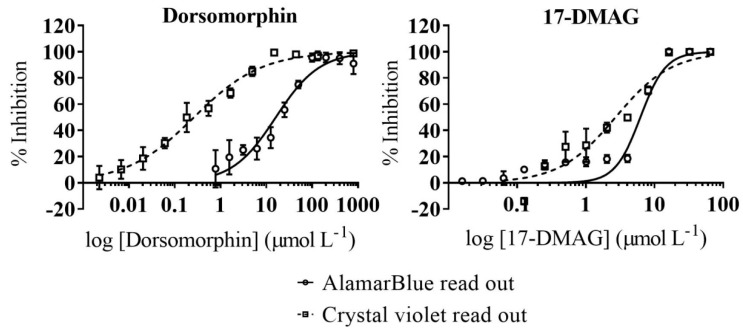
Dose–Response Curves of Vero Cell Toxicity Assays. AlamarBlue readout is represented by empty circles and straight line while crystal violet read out by empty squares and dashed line. Graphs represent mean results and SD of at least three biological replicas.

**Figure 4 ijms-22-00688-f004:**
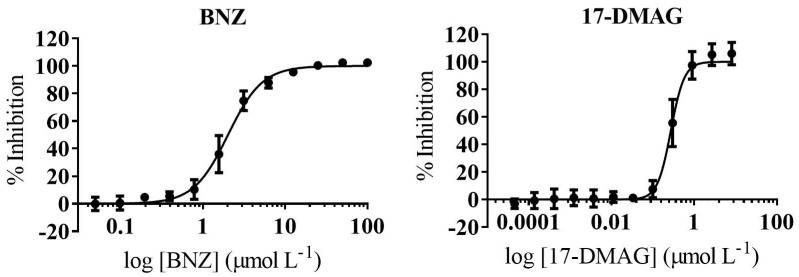
Amastigote-Specific Activity Curves of BNZ and 17-DMAG. Graphs represent mean results and SD of at least three biological replicas.

**Figure 5 ijms-22-00688-f005:**
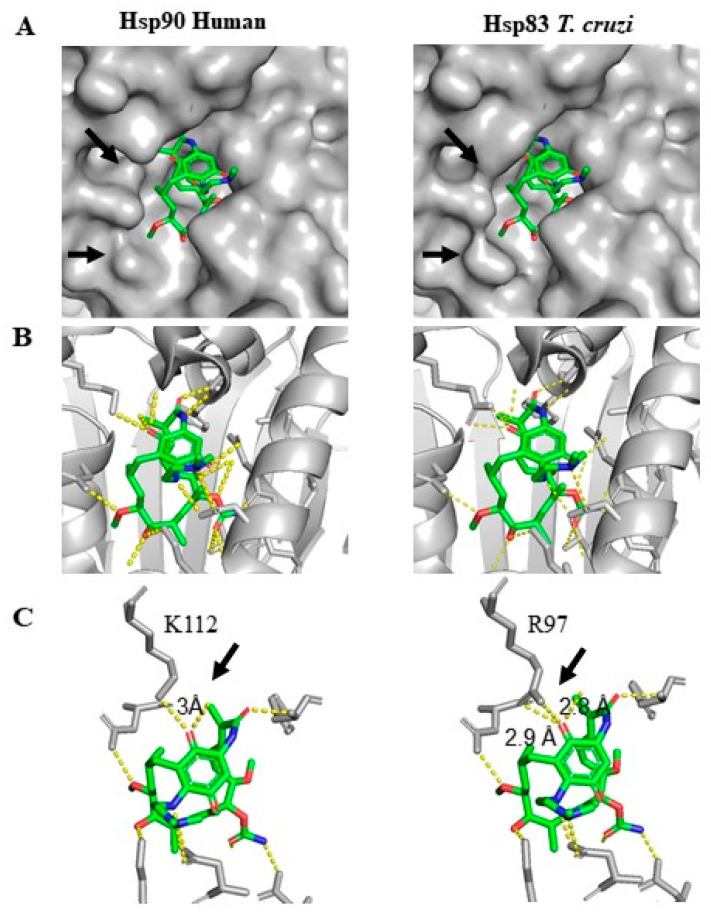
Binding of 17-DMAG to Human Hsp90 and *T. cruzi* Hsp83. (**A**) Binding pocket surface of Hsp90 (**left**) and Hsp83 (**right**) with 17-DMAG. (**B**) Zoom-in to the residue interactions of Hsp90 (**left**) and Hsp83 (**right**) with 17-DMAG. (**C**) Zoom-in to the K112 residue of Hsp90 (**left**) and R97 residue of Hsp83 (**right**) interaction with 17-DMAG. PyMol software was used for visualizing PDB ref.: 1OSF [32]. 17-DMAG structure is represented in green and protein structure in grey. In panel A, arrows indicate morphological differences between both surfaces. In panel C, arrows show the position of expected H-bonds between 17-DMAG and K112 or R97 residues, respectively. Hsp90/83 from panel B is represented as cartoons; in panel C they are represented as sticks [33].

**Figure 6 ijms-22-00688-f006:**
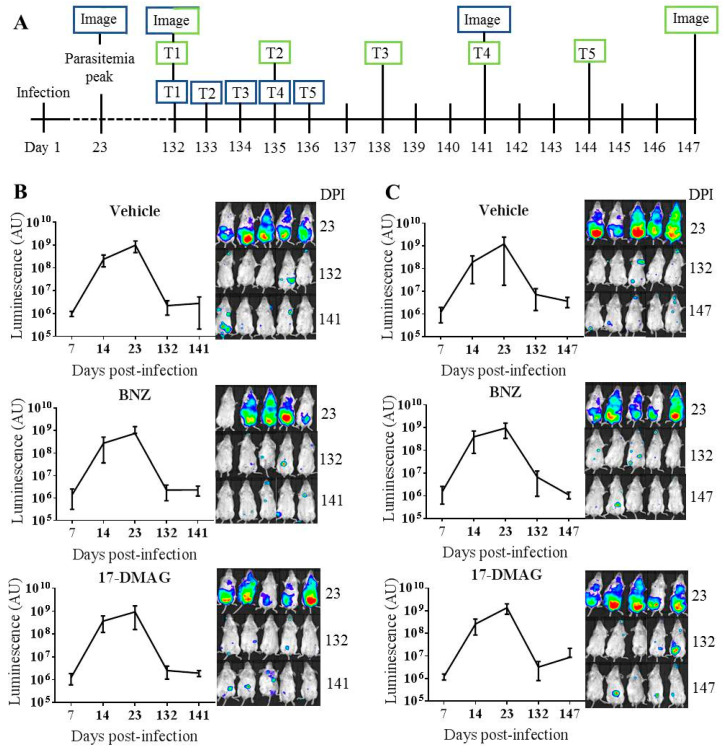
In vivo model of chronic *T. cruzi* infection. (**A**) Experimental timeline of events. Blue squares refer to experimental groups treated daily for five days whereas green squares to groups treated five times over 15 days. (**B**) Parasitemia evolution represented by graphs and images of animals treated daily for five days. (**C**) Parasitemia evolution represented by graphs and images of animals treated five times over 15 days. The luminescence reactivity shown longitudinally in panels B and C are mean and SD values of all the mice within each corresponding group.

**Table 1 ijms-22-00688-t001:** Table Displaying the Averaged IC_50_, TC_50_, and SI Values Obtained in the Study.

Compound	IC_50_	IC_50_ ^a^	AlamarBlue Assay			Crystal Violet Assay		
			TC_50_	SI	SI ^a^	TC_50_	SI	SI ^a^
**BNZ**	1.63	2.02	243.8	149.6	120.6	140.2	86.0	69.4
**Dorsomorphin**	0.24	N.T.	16.6	69.2	-	0.26	1.2	-
**17-DMAG**	0.017	0.27	6.23	366.5	23.1	2.97	174.7	11.0

IC_50_ and TC_50_ values are, respectively, defined as the compound concentration that inhibits parasite growth by 50% or is toxic to 50% of cells. SI stands for selectivity index and is determined by the rate between TC_50_ and IC_50_ values. ^a^, Antiamastigote assay; all IC_50_ and TC_50_ values are shown as µmol L^−1^; N.T., not tested.

## Data Availability

The data presented in this study are available in the article or Appendix A.

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
