# Peer review of "Anti-Trypanosoma cruzi Activity of Metabolism Modifier Compounds"

_ijms, 2021, doi:10.3390/ijms22020688_

Round 1
Reviewer 1 Report
Title :
Anti-Trypanosoma cruzi activity of metabolism modifier compounds
Authors :
Nieves Martinez-Peinado, Clara Martori, Nuria Cortes-Serra, Julian Sherman, Ana Rodriguez, Joaquim Gascon, Jordi Alberola, Maria-Jesus Pinazo Alheli, Rodriguez-Cortes and Julio Alonso-Padilla
Chagas disease is a main public health problem worldwide. The current drugs (benznidazole (BNZ) and nifurtimox (NFX)) are inefficient during the chronic phase, so the identification and the development of new drugs to treat it remain a priority.
The new drugs will be targeting fundamental enzymes or metabolic pathways either in the parasite or in the host could be a source of potential anti-parasitic treatments. As mentioned by the authors and others host metabolic networks of energy production, nucleotide metabolism, pteridine biosynthesis, and fatty acid oxidation can be targeted. Authors evaluated several candidate coumpounds (AICAR, dorsomorphin, SC79, Akti-1/2, oligomycin, etomoxir, 17-DMAG, doxycycline and sodium salicylate).
First of all, authors have test the anti T. cruzi activity of their candidates compared to benznidazole. 6/9 of their candidates were inactive against the parasite. However, dorsomorphin, oligomycin and 17-DMAG showed an inhibition activity more important than BNZ. Their IC50 values were lower than the BNZ one.
Then authors test the cell toxicity to confirm if their candidates have an anti-T. cruzi assay specific against the parasite cells. Only two compounds remain in the race (dorsomorphin, oligomycin and 17-DMAG).
Vero cells toxicity assay of dorsomorphin leads to exclusion of this candidate whereas 17-DMAG remains in the race. Then, authors test 17-DMAG on amastigotes. They found that the anti-amastigote activity of 17-DMAG is poor or at least modest.
Based on in silico analysis, authors have shown that 17-DMAG 17-DMAG might bind T. cruzi Hsp90 homologue Hsp83 with good affinity. So, authors finally test 17-DMAG in a mouse model. In this model infection reached a peak at 32 dpi and treatment started at 132 dpi for 5 consecutive days or five times every three days. Unfortunately, none of the groups treated with 17-DMAG showed statistically significant differences versus either vehicle or BNZ groups (based on luciferase assay)
Comments:
- In their introduction, authors indicated biological processes that can targeted during drugs development. Authors cited the work done previously by Caradonna et al. I clearly invite the authors to extend this section, as a mini review, as their candidate drug screening results didn’t reach completely their expectations.
- Among these candidate biological processes, it will be nice to add the mitochondria authors may provide a discussion about it.
- In their results authors used several indicators such as SI, TC50, IC50. Even I some of them are obvious for readers these ones must be clearly defined.
- Authors used as selective index (SI; TC50 / IC50 ratio) > 10 without discussing these indexes.
- In the mouse model, the efficiency of 17-DMAG was investigate by a luciferase assay. The luciferase signal is proportional to the cruzi load, and it is considered a surrogate for parasitemia. I have a major concern on the way to conduct this analyse. The goal of the experiments was to identify a drug efficient for chronic phase and in their mouse model authors look only for parasitemia. Is it not possible to conduct assays monitoring the chronic phase (fibrosis, ECG…) I was surprised that authors didn’t conduct assay on Hsp activity too.
- My second concern is about the starting point for treatment. For me, this one is too late.
- It is too pitty that the authors did get a group of mice receiving simultaneously 17-DMAG and BNZ.
Author Response
Comments and Suggestions for Authors:
Chagas disease is a main public health problem worldwide. The current drugs (benznidazole (BNZ) and nifurtimox (NFX)) are inefficient during the chronic phase, so the identification and the development of new drugs to treat it remain a priority.
The new drugs will be targeting fundamental enzymes or metabolic pathways either in the parasite or in the host could be a source of potential anti-parasitic treatments. As mentioned by the authors and others host metabolic networks of energy production, nucleotide metabolism, pteridine biosynthesis, and fatty acid oxidation can be targeted. Authors evaluated several candidate compounds (AICAR, dorsomorphin, SC79, Akti-1/2, oligomycin, etomoxir, 17-DMAG, doxycycline and sodium salicylate).
First of all, authors have test the anti T. cruzi activity of their candidates compared to benznidazole. 6/9 of their candidates were inactive against the parasite. However, dorsomorphin, oligomycin and 17-DMAG showed an inhibition activity more important than BNZ. Their IC50 values were lower than the BNZ one.
Then authors test the cell toxicity to confirm if their candidates have an anti-T. cruzi assay specific against the parasite cells. Only two compounds remain in the race (dorsomorphin, oligomycin and 17-DMAG).
Vero cells toxicity assay of dorsomorphin leads to exclusion of this candidate whereas 17-DMAG remains in the race. Then, authors test 17-DMAG on amastigotes. They found that the anti-amastigote activity of 17-DMAG is poor or at least modest.
Based on in silico analysis, authors have shown that 17-DMAG might bind T. cruzi Hsp90 homologue Hsp83 with good affinity. So, authors finally test 17-DMAG in a mouse model. In this model infection reached a peak at 32 dpi and treatment started at 132 dpi for 5 consecutive days or five times every three days. Unfortunately, none of the groups treated with 17-DMAG showed statistically significant differences versus either vehicle or BNZ groups (based on luciferase assay).
Comments:
- In their introduction, authors indicated biological processes that can targeted during drugs development. Authors cited the work done previously by Caradonna et al. I clearly invite the authors to extend this section, as a mini review, as their candidate drug screening results didn’t reach completely their expectations. We thank reviewer´s comments, and this section has now been extended as suggested. Please see lines 66-74.
- Among these candidate biological processes, it will be nice to add the mitochondria authors may provide a discussion about it. Following reviewer comment we have added the mitochondria role in lines 93-94.
- In their results authors used several indicators such as SI, TC50, IC50. Even if some of them are obvious for readers these ones must be clearly defined. We thank reviewer for this suggestion, and have added definitions for IC50 and TC50 in Material and Methods section 4.8 (see lines 510-511). Please, see line 161 where SI is defined. Besides, IC50, TC50 and SI are also explained in Table 1 footnote caption.
- Authors used as selective index (SI; TC50 / IC50 ratio) > 10 without discussing these indexes. We have now added a referenced explanation in lines 162-163.
- In the mouse model, the efficiency of 17-DMAG was investigated by a luciferase assay. The luciferase signal is proportional to the T. cruzi load, and it is considered a surrogate for parasitemia. I have a major concern on the way to conduct this analyse. The goal of the experiments was to identify a drug efficient for chronic phase and in their mouse model authors look only for parasitemia. Is it not possible to conduct assays monitoring the chronic phase (fibrosis, ECG…) I was surprised that authors didn’t conduct assay on Hsp activity too. This in vivo model has been previously used for the evaluation of drugs against T. cruzi infections and the correspondence between luciferase activity and the number of parasites determined (please see PMID: 21912715 or PMID: 24144412). In agreement with reviewer, it would have been very interesting to monitor fibrosis, ECG and so on. Unfortunately, we did not have the means to perform them in the BSL-2 animal facility that is required for T. cruzi infected mice. We expect to secure funds to run analysis of Hsp enzymes target inhibition, which we would have to clone and express, but we cannot do this at present.
- My second concern is about the starting point for treatment. For me, this one is too late. The reason for the late treatment is to ensure that animals are in the chronic phase of infection, which occurs only after about 100 days after infection. This model has been previously established (see for example PMID: 23945371 or PMID: 26014936). Current need for Chagas disease drugs is mainly for the treatment of the chronic disease, which is when most patients are diagnosed. At this stage, parasitemia is low and parasites hide deep into the tissues. There they are difficult to reach, which compromises drug efficacy.
- It is too pitty that the authors did get a group of mice receiving simultaneously 17-DMAG and BNZ. We agree with reviewer and did think about co-administration of both drugs. However, the number of animals was limited, same as our funds to perform the chronic in vivo experiments. These are particularly expensive since animals must be housed for a long period of time.
Reviewer 2 Report
In this work, Martinez-Peinado et al. studied the activity of nine compounds, with known activity over the cellular metabolism, as potential candidates (first in vitro then in vivo) to treat the infection caused by protozoan parasite Trypanosoma cruzi. Currently there are only two valid approved drugs to treat this parasitic infection. However, both their moderate to severe adverse effects and their limited effectivity have urged the discovery of new drugs to fight against this neglected disease. This task has been unsuccessful for decades until today, including several big failures in clinical trials. Therefore, the complexity of this objective and the negative results inferred along the full manuscript confirm once again that an effective and safe treatment against Chagas disease are urgently needed.
In addition to the negative results, the overall contribution to the field is very scarce. As expected, the selection of these compounds showed a high toxicity to be used as an in vivo treatment for a chronic infection that requires long periods of treatment. The use of English in the full manuscript is very poor with many mistakes, to the point that makes it very difficult to understand the content. Even sections such the M&M, where it should be as easy - as reading a protocol to follow - there are difficulties to figure out what exactly has been done. A major review of this point is urgently required by a native English speaker, who should proof-read the manuscript before further submission. Major issues in the content are continuously detected along the manuscript, precisely:
- The headline is missing (line 99?)
- Sections 2.1, 2.2 and 2.3 should be merged and better explained. In section 2.1 the reader must infer which form of the parasite (epimastigotes, amastigotes, trypomastigotes) are assayed. It is not until the method section, when retrospectively you can figure out that both cells and parasites were added at the same time. Therefore, the compound could have affected to the attachment of the cells. Hence the term of “growth inhibition” (line 109) doesn't not apply here.
- In Table 1 is chaotic to follow all the IC50s, TC50s, SIs on the top row, despite the super-indexes. This need to be clarified.
- Including oligomycin in the study is not appropriate, even the authors mention (lines 127 -129) there are serious issues about its toxicity. Testing “poisons” to show they kill T. cruzi but after you cannot use them to treat the disease is a very poor hypothesis.
- Lines 131 – 134 authors talk about two methods for measure cytotoxicity. The explanation of this methods would be more appropriate in the M&M section. Furthermore, the second method in neither well explained in neither Results or M&M sections. What crystal violet method measures?
- Like in point 4) the lines 158 – 159 can be deleted.
- Lines 160 – 161 are more M&M material rather than results themselves.
- Lines 162 – 166 are unclear, they should define potency and SI before use the terms. A better explanation between the drugs activity and the forms of the parasite assayed should be included.
- Authors use a different strain for the in vivo study compared to the in vitro Considering the huge genetic diversity of T. cruzi, they should give some data about the new strain, specially regarding to drug sensitivity (eg BZN or NFX – as reference drugs).
- Lines 209 to 223 are very vague explained, mixing results with M&M sections. Lines 212 – 213 the dosing schedule is unclear. Line 223 the use “healthy mouse” for the control, aren’t healthy the mice for the experiment?
- Figure 6A, Mice are imaged on day 1 of Treatment instead of before treatment?
- Lines 231 – 233 are a bit conflictive, as during the chronic phase of T. cruzi infection in mice it has been reported that the fluctuation in the parasitemia detected using bioluminescent models are common. So, to assume that the trend is towards reduction because of the drug is complicated. Line 235 – “poorly infected” is a very vague term.
- Lines 248 – 250 give a very risky assumption as the statement has not been proven in the manuscript.
- Lines 272 – 295 should be shortened and moved to introduction section, as this is a review of 17-DMAG.
- Lines 333 – 357 summarise the negative results of the study and the impossibility to extend dosage or length of treatment with the “best” drug of the study.
- Line 360, compounds need to be capitalised to follow the format of the headlines and subheadlines.
- Lines 374 and 380, P-S must be expressed in units/ml or mg/ml, percentage is not reproducible, as otherwise it would be inferred 1mg per 100ml.
- Line 393 – After for?
- Lines 398 – 402 need to be clarified. Why to infect first and then de-attach the cells? Why not infect with a particular MOI then wash?
- Line 426. Triplicate mean: 3 wells per condition or 3 independent experiments of 3 wells?
- Lines 484 – 492, institutions names and departments should be homogenized and be all provided in English or in the native language, but not a mix of them. Line 489 - Ministry of Health doesn't exist for this institution (please use Council of Health or Conselleria de Sanitat).
Author Response
Comments and Suggestions for Authors:
In this work, Martinez-Peinado et al. studied the activity of nine compounds, with known activity over the cellular metabolism, as potential candidates (first in vitro then in vivo) to treat the infection caused by protozoan parasite Trypanosoma cruzi. Currently there are only two valid approved drugs to treat this parasitic infection. However, both their moderate to severe adverse effects and their limited effectivity have urged the discovery of new drugs to fight against this neglected disease. This task has been unsuccessful for decades until today, including several big failures in clinical trials. Therefore, the complexity of this objective and the negative results inferred along the full manuscript confirm once again that an effective and safe treatment against Chagas disease are urgently needed.
In addition to the negative results, the overall contribution to the field is very scarce. As expected, the selection of these compounds showed a high toxicity to be used as an in vivo treatment for a chronic infection that requires long periods of treatment. The use of English in the full manuscript is very poor with many mistakes, to the point that makes it very difficult to understand the content. Even sections such the M&M, where it should be as easy - as reading a protocol to follow - there are difficulties to figure out what exactly has been done. A major review of this point is urgently required by a native English speaker, who should proof-read the manuscript before further submission. Major issues in the content are continuously detected along the manuscript, precisely:
We thank the reviewer´s comments and suggestions, and proceed to answer her/his concerns in blue colour below. In addition, the use of English has been revised in the entire manuscript as indicated.
- The headline is missing (line 99?). Following reviewer suggestion we have now added the “Results” headline.
- Sections 2.1, 2.2 and 2.3 should be merged and better explained. In section 2.1 the reader must infer which form of the parasite (epimastigotes, amastigotes, trypomastigotes) are assayed. It is not until the method section, when retrospectively you can figure out that both cells and parasites were added at the same time. Therefore, the compound could have affected to the attachment of the cells. Hence the term of “growth inhibition” (line 109) doesn't not apply here. We have now added which forms of the parasite (mammalian stages) are targeted by the anti-parasitic assay, and “growth inhibition” term was changed for “activity against” them (see line 119). If it cannot be discarded that compounds may affect to the attachment of the cells in our primary anti-parasitic assay, we usually check that cells correctly attached and properly formed monolayers by microscopy visualization. Besides, upon obtaining anti-parasitic assay results, all compounds that are suspected to have anti-parasitic activity are analyzed in a secondary Vero cell toxicity assay. If a compound/drug appears to be toxic to them, it is not specific against the parasite, flagged as false positive, and discarded from further evaluation. Regarding fusion of sections, we would rather prefer to leave as they are. Otherwise section would be quite long. New explanations have been included for clarity purposes in the footnote caption of Table 1, in section 2.2 (see lines 147-163), and in section 2.3 (see lines 181-188).
- In Table 1 is chaotic to follow all the IC50s, TC50s, SIs on the top row, despite the super-indexes. This need to be clarified. We thank the reviewer for her/his suggestion and have now added new headings within the table for clarity purposes. Plus terms IC50, TC50 and SI are defined in its footnote.
- Including oligomycin in the study is not appropriate, even the authors mention (lines 127 -129) there are serious issues about its toxicity. Testing “poisons” to show they kill T. cruzi but after you cannot use them to treat the disease is a very poor hypothesis. Our objective was to find out what was the effect on the parasites when the host cell mitochondria were blocked. Many studies on the role of immune-metabolism include oligomycin in their functional assays (e.g. see PMID: 31841511). This feature has now been included in the text (see lines 93-94), and for clarity purposes we have moved oligmycin results to supplementary material (Figure S2_v2) as indicated.
- Lines 131 – 134 authors talk about two methods for measure cytotoxicity. The explanation of this methods would be more appropriate in the M&M section. Furthermore, the second method in neither well explained in neither Results or M&M sections. What crystal violet method measures? An improved explanation for crystal violet method is now provided for clarity purposes in “Materials and Methods” section 4.5 (see lines 454-459).
- Like in point 4) the lines 158 – 159 can be deleted. Those lines have been now deleted.
- Lines 160 – 161 are more M&M material rather than results themselves. We thank the reviewer for this comment, but, same as it happens with other features of the manuscript, the fact that “Materials and Methods” section is located at the end of the document may preclude readers from having a clearer understanding of the text. Thus, it is for clarity purposes that we rather prefer to leave that sentence where it is to highlight that we are really targeting the amastigote forms.
- Lines 162 – 166 are unclear, they should define potency and SI before use the terms. A better explanation between the drugs activity and the forms of the parasite assayed should be included. We have now added the term IC50 where corresponding for clarity purposes. Besides, an explanation on the drugs activity and the parasite forms impacted is also included. SI definition is detailed in line 161.
- Authors use a different strain for the in vivo study compared to the in vitro. Considering the huge genetic diversity of T. cruzi, they should give some data about the new strain, specially regarding to drug sensitivity (eg BZN or NFX – as reference drugs). The use of two different parasite types respectively in the in vitro and in vivo assays is an added value indeed. Tulahuen strain belongs to DTU VI, while Brazil is DTU I. Tulahuen recombinant parasites expressing beta-galactosidase have been widely used for in vitro anti-T. cruzi drug discovery and shown a robust performance and stable drug susceptibility. Brazil strain has also been shown to be susceptible to BNZ, yet same as Tulahuen having high virulence (see Table 2 in Minning et al., BMC Genomics 2011). That reference has now been included in the manuscript (ref. 58), and a brief explanation is provided in lines 490–491.
- Lines 209 to 223 are very vague explained, mixing results with M&M sections. Lines 212 – 213 the dosing schedule is unclear. Line 223 the use “healthy mouse” for the control, aren’t healthy the mice for the experiment? Considering that Int J Mol Sci sections organization entail that “Materials and Methods” go at the end of the manuscript, we provide a brief explanation of most relevant “MMs” features in “Results” section entries for clearer reading. With the term “healthy mouse” we meant non-infected. We thank the reviewer for this comment and changed “healthy” for “non-infected”.
- Figure 6A, Mice are imaged on day 1 of Treatment instead of before treatment? Animals are always imaged before treatment administration in order to establish groups that have similar parasite loads.
- Lines 231 – 233 are a bit conflictive, as during the chronic phase of T. cruzi infection in mice it has been reported that the fluctuation in the parasitemia detected using bioluminescent models are common. So, to assume that the trend is towards reduction because of the drug is complicated. Line 235 – “poorly infected” is a very vague term. It is true that fluctuation in the parasitemia using bioluminescent models may occur, but considering that we use a group of 5 mice, reported values are consistent so as to attribute the observed average reduction to the effect of the drug. We have changed the words “poorly infected” for “less infected”.
- Lines 248 – 250 give a very risky assumption as the statement has not been proven in the manuscript. We have now re-worded that sentence.
- Lines 272 – 295 should be shortened and moved to introduction section, as this is a review of 17-DMAG. We appreciate reviewer comment and have shortened the paragraph, but we would prefer to leave those lines in the “Discussion” section.
- Lines 333 – 357 summarise the negative results of the study and the impossibility to extend dosage or length of treatment with the “best” drug of the study. That is it, unfortunately. At least it would be for the daily administration of 17-DMAG. Whether longer intermittent regime can provide improved results will have to be evaluated.
- Line 360, compounds need to be capitalised to follow the format of the headlines and subheadlines. We have now corrected this issue. We thank the reviewer for this comment, and also capitalized words in the subheadings of “Results” section to follow the journal style.
- Lines 374 and 380, P-S must be expressed in units/ml or mg/ml, percentage is not reproducible, as otherwise it would be inferred 1mg per 100ml. We have added “100 units/ml of penicillin and 100 μg/ml of streptomycin; P-S” in lines 410-411.
- Line 393 – After for? We have deleted “for”.
- Lines 398 – 402 need to be clarified. Why to infect first and then de-attach the cells? Why not infect with a particular MOI then wash? In our experience, there are no major differences in performing both methods in terms of retrieved IC50 values. However, infecting in-bulk is operationally advantageous. Moreover, by infecting the cells directly in the flasks and then detaching them to run the assays we avoid disturbing the monolayers with extra washing steps performed with the multi-channel pipette, which may introduce assay variability.
- Line 426. Triplicate mean: 3 wells per condition or 3 independent experiments of 3 wells? It refers to three biological replicas, i.e. three independent experiments. In all cases, values provided are means of at least three independent experiments.
21. Lines 484 – 492, institutions names and departments should be homogenized and be all provided in English or in the native language, but not a mix of them. Line 489 - Ministry of Health doesn't exist for this institution (please use Council of Health or Conselleria de Sanitat). We thank the reviewer for this suggestion. We have homogenized and corrected funders´ names.
Reviewer 3 Report
Dear Editor,
I checked the manuscript titled: “Anti-Trypanosoma cruzi activity of metabolism modifier compounds” which I found interesting and novelty, and I believe it could be accepted for publication in International Journal of Molecular Sciences, after minor changes.
Authors are checking the activity of nine bio-energetic compound, among them they are checking also the oligomycin, previously described highly toxic for human. Based on this (page 4 line 128) they decided to not further pursue the experiments on this compound. It is interesting to observe the growth inhibition effect of oligomycin on T. cruzi, but I would prefer to refer to these results in the text either than present them in Figure 2. I would suggest to authors to add in Fig. 2 the results of control drug BZN and to present the oligomycin results as supplementary.
Authors are presenting in Figures 3 and S2, both methods, AlamarBlue and Crystal Violet. Authors should lightening the graphs. I would suggest using smaller symbols and lines. I would also suggest not using black fulfilled symbols. Moreover, a legend showing the symbols for each method would be better than write it.
The section “2.4. Hsp90 is the target of 17-DMAG” in my opinion should be as the last one, after the section “2.5. In vivo anti-T. cruzi activity of 17-DMAG in a mouse model of chronic infection.”
Figure 6 should be improved. It would be better to present just the mean with SD instead of the five mouse results plus the mean with the SD. As it is presented now is not easy to read. Also, please be careful about “C” graphs, two of them are ending at day 141 instead of day 147.
Discussion, lines 247 – 265: there are redundant information that I would rather summarize and focus on the real discussion.
Author Response
Comments and Suggestions for Authors:
Dear Editor,
I checked the manuscript titled: “Anti-Trypanosoma cruzi activity of metabolism modifier compounds” which I found interesting and novelty, and I believe it could be accepted for publication in International Journal of Molecular Sciences, after minor changes.
Authors are checking the activity of nine bio-energetic compound, among them they are checking also the oligomycin, previously described highly toxic for human. Based on this (page 4 line 128) they decided to not further pursue the experiments on this compound. It is interesting to observe the growth inhibition effect of oligomycin on T. cruzi, but I would prefer to refer to these results in the text either than present them in Figure 2. I would suggest to authors to add in Fig. 2 the results of control drug BZN and to present the oligomycin results as supplementary. We thank reviewer´s comments. Following her/his suggestion we show oligomycin results in the text and now present them as supplementary figure (Figure S2_v2). BNZ curve has been included in figure 2 (uploaded now as Figure 2_v2). Please note that former supplementary figure 3 will be now Figure S4.
Authors are presenting in Figures 3 and S2, both methods, AlamarBlue and Crystal Violet. Authors should lightening the graphs. I would suggest using smaller symbols and lines. I would also suggest not using black fulfilled symbols. Moreover, a legend showing the symbols for each method would be better than write it. We have now corrected figures´ layout for clarity purposes. New v2 items are provided for each of them. Please note that former supplementary figure 2 y now Figure S3.
The section “2.4. Hsp90 is the target of 17-DMAG” in my opinion should be as the last one, after the section “2.5. In vivo anti-T. cruzi activity of 17-DMAG in a mouse model of chronic infection.” We appreciate the reviewer´s suggestion but we would rather prefer to keep the order of the sections as it is now, so that all evidences obtained in vitro and in silico support the move to evaluate 17-DMAG in vivo.
Figure 6 should be improved. It would be better to present just the mean with SD instead of the five mouse results plus the mean with the SD. As it is presented now is not easy to read. Also, please be careful about “C” graphs, two of them are ending at day 141 instead of day 147. Following the reviewer suggestion, we have now changed the graph and provide intensified mean and SD lines. Also, we changed panel “C” graphs and they are now ending at day 147. Thank you for this observation.
Discussion, lines 247 – 265: there are redundant information that I would rather summarize and focus on the real discussion. This feature has now been corrected and redundant information deleted.
Round 2
Reviewer 1 Report
thanks for the responses to all my queries
Reviewer 2 Report
No further comments required, thank you to the authors for having in consideration some of the suggestions and changes.